

# Multiple time-scale variations of fronts in the Seto Inland Sea, Japan

Menghong Dong[1], Xinyu Guo[2*]

[1]State Key Laboratory of Satellite Ocean Environment Dynamics, Second Institute of Oceanography, Ministry of Natural Resources, Hangzhou, 310012, China
[2]Center for Marine Environmental Studies, Ehime University, Matsuyama, 7908577, Japan

*Correspondence to*: Xinyu Guo (guo.xinyu.mz@ehime-u.ac.jp), Orcid ID: 0000-0002-4832-8625

**Abstract.** The Seto Inland Sea (SIS) is a critical semi-enclosed coastal sea in Japan, characterized by intricate coastlines and narrow straits that give rise to various fronts. Despite extensive research on tidal fronts, knowledge gaps persist regarding their spatiotemporal dynamics, particularly in certain poorly documented regions. Additionally, the understanding of
thermohaline fronts, which emerge during winter, requires further investigation. We aimed to enhance our understanding of tidal and thermohaline fronts in the SIS by analyzing their dynamic processes, including intra-tidal and spring-neap tidal cycles, seasonal variations, and anomalous frontal variability. Using a gradient-based algorithm with an advanced contextual feature-preserving median filter, we processed the high-resolution sea surface temperature dataset to detect and quantify tidal and thermohaline fronts. Our analysis revealed the presence of numerous tidal fronts, predominantly influenced by the $M_2$
tide, across the SIS, with substantial variations in spatial amplitude due to complex coastlines and narrow straits. Intra-tidal movements of tidal fronts corresponded to ebb and flood currents, while spring-neap tidal cycles and seasonal shifts influenced frontal positions and intensities. Additionally, thermohaline fronts were identified in certain regions during winter, characterized by large horizontal temperature and salinity gradients. This study enhances the understanding of tidal and thermohaline fronts in the SIS, emphasizing the importance of intra-tidal and wind-driven influences on frontal dynamics.
However, limited observational coverage and resolution emphasize the need for further research to explore long-term temporal changes and better grasp the influence of ambient currents and wind patterns. Such insights are vital for effective coastal management and environmental monitoring in the SIS region.

## 1 Introduction

The Seto Inland Sea (SIS) is Japan's largest semi-enclosed coastal sea, surrounded by the Honshu, Shikoku, and Kyushu
Islands. It connects to the Pacific Ocean through the Kii Channel to the east and the Bungo Channel to the west (Figure 1a). It has a length of ~500 km, width of 4–50 km, and an average depth of 40 m. It is interspersed with ~600 small islands that form complex coastlines and many narrow straits with strong tidal currents. In addition to the islands and straits, the wide basins inside are called various "nada" in Japanese (Figure 1a). This complicated geometry results in large variations in the marine environment, giving rise to various fronts within the SIS. Fronts, the surface convergence zones between two water
masses (Yanagi, 1987), are crucial in material transport and biological processes.

Many fronts in the SIS are tidal fronts, which are defined as transition zones between stratified water in summers due to surface heating and vertically mixed water due to tidal stirring (Dong and Guo, 2021; Sun and Cho, 2010; Sun and Isobe, 2006; Takeoka, 1990, 2002; Takeoka et al., 1993, 1997; Yanagi et al., 1992; Yanagi, 1980, 1987; Yanagi and Okada, 1993). In the SIS, four major tidal constituents ($M_2$, $S_2$, $K_1$, and $O_1$) are present, of which $M_2$ tide is predominant (Guo et al., 2013;
Higo et al., 1980; Yanagi and Higuchi, 1981). Tides from the Pacific Ocean enter the SIS through the Bungo and Kii Channels, propagating in opposing directions and converging at Hiuchi-nada (Figure 1a), resulting in the highest tide amplitude in this basin. Tidal currents within the SIS exhibit substantial spatial amplitude variations owing to their complex coastlines and narrow straits. While tidal currents are very weak in basins like Hiuchi-nada (0.1 m/s amplitude), they can reach speeds of up to 5 m/s in narrow straits such as the Kurushima Strait and Naruto Strait (Figure 1a) (Guo et al., 2013).





Considerable spatial variations in the tidal current amplitudes around the narrow straits facilitate tidal front formation. According to Simpson and Hunter's energetics (Simpson and Hunter, 1974), tidal fronts align along contours of a critical value of $\log_{10}(h/u^3)$, with the critical value in the SIS ranging from 2.5 to 3.0 (Yanagi and Okada, 1993). Here $h$ is the water depth, and $u$ is the tidal current amplitude.

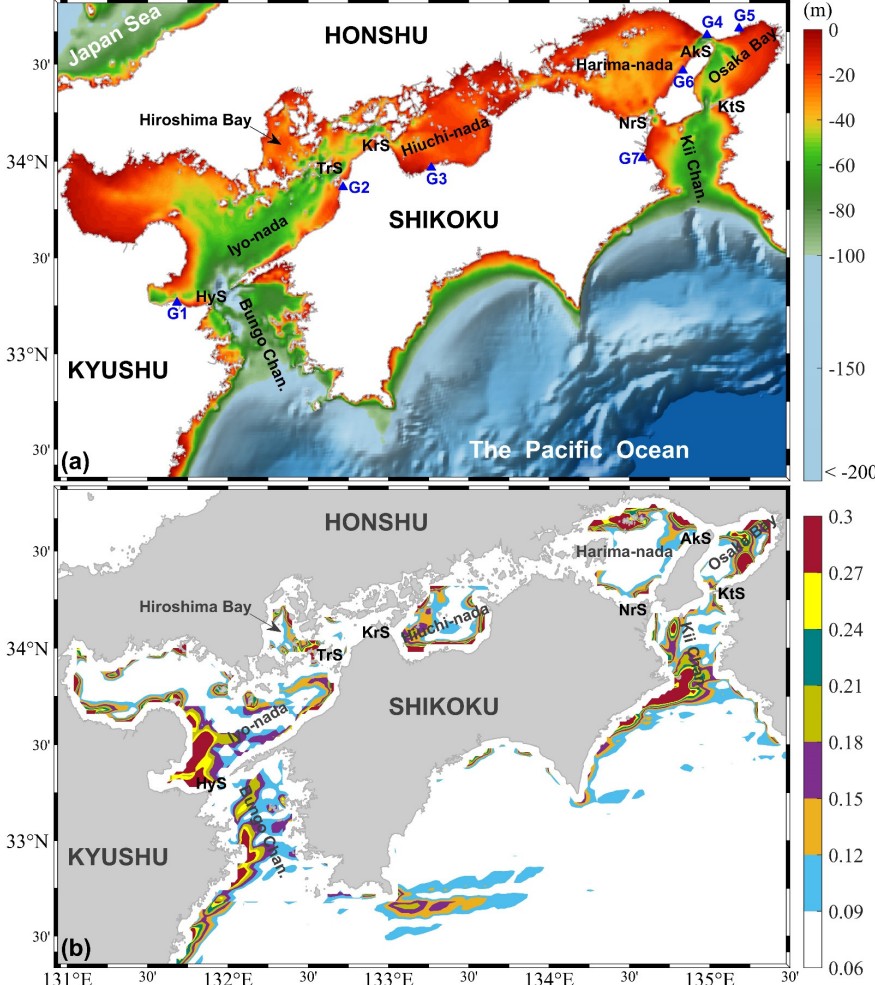

**Figure 1**. (a) Location and bathymetry of the SIS. The blue triangles indicate the gauge stations. 'HyS,' 'TrS,' 'KrS,' 'AkS,' 'NrS,' and 'KtS' denote Hayasui Strait, Tsurushima Strait, Kurushima Strait, Akashi Strait, Naruto Strait, and Kantan Strait, respectively. (b) The appearance frequency of the fronts in the SIS.

Before the availability of high spatial and temporal resolutions observational data, research on tidal fronts primarily focused
on their formation and variations over inter-tidal periods, including the spring-neap tide cycle and seasonal variations in front position and intensity. Tidal fronts in various regions of the SIS, such as Bungo Channel (Yanagi et al., 1992; Yanagi and Koike, 1987; Yanagi and Okada, 1993; Yanagi and Tamaru, 1990), Iyo-nada (Sun and Isobe, 2006; Takeoka et al., 1993, 1997), Hiuchi-nada (Takeoka, 1990), and Osaka Bay (Yanagi and Takahashi, 1988), have been investigated through observational data and numerical simulations. For example, Yanagi and Tamaru (1990) observed minimal seasonal
variations in the tidal front position in the Bungo Channel. However, they noted substantial intensity fluctuations from April to August due to the considerable changes in heating rates during this period. During the spring-neap tide cycle, the front



tends to be closer to the well-mixed water area in neap tide than in spring tide, with intensity increasing from neap tide to spring tide (Yanagi and Tamaru, 1990). However, certain tidal fronts near narrow straits in the SIS remain poorly documented, including those on the western side of Tsurushima Strait and the northern side of Akashi Strait (Figure 1b).

Since the magnitude of $h/u^3$ represents the mean position of the tidal front over a tidal period, it is often used for studying inter-tidal variations in the fronts. However, the tidal currents move the instantaneous frontal position within a tidal cycle. Dong and Guo (2021) confirmed the intra-tidal movement of a tidal front in the Bungo Channel using five-year high-resolution satellite Sea Surface Temperature (SST) data (hourly in temporal and 2 km in spatial resolution). They demonstrated that this movement is mainly controlled by tidal current advection. During a semidiurnal tidal cycle, the intra-

tidal movement of the front corresponds to ebb and flood currents. Additionally, the intensity of the front increases during the ebb current phase, which carries the front toward the stratified area, but decreases during the flood current phase, which drives the front in the opposite direction (Dong and Guo, 2021).

In addition to intra-tidal variations, Dong and Guo (2021) reported that the basin-scale residual current affects spring-neap tidal variations in the front position. In April, the residual current in the SIS was weak, causing the front to be closer to the

stratified water area during spring tide because the tidal stirring power was larger during spring tide than during neap tide. However, the residual current strengthened in July due to surface heating and river water input. The residual current also varied with a spring-neap tidal cycle and was intensigied during the neap tide but weakened during the spring tide (Kuo-Chuin Wong, 1994). Since the residual current flows from the mixing area to the stratified area, it drives the front closer to the stratified area during neap tide than during spring tide.

The significant intra- and month-dependent variations observed in the tidal front of the Bungo Channel (Dong and Guo, 2021) prompt the question of whether similar variations occur in tidal fronts elsewhere in the SIS. Moreover, understanding their long-term temporal changes is challenging without a comprehensive description of intra-tidal front variations due to potential undersampling in observations. Wind-driven currents also affect tidal fronts. Wang et al. (1990) reported that the wind-driven advection of well-mixed waters over warmer stratified waters causes convective instability and intensifies the

tidal front in the Irish Sea. The elevated wind speeds in the SIS during summer (Shi et al., 2011) necessitate consideration of their influence on tidal fronts in the region.

In addition to the tidal fronts that emerge around the straits in the SIS during summer, another type of front known as the thermohaline front appears during winter in certain areas of the SIS, such as the western part of Iyo-nada and the southern entrance of the Kii Channel (Figure 1). The formation of thermohaline fronts depends on surface cooling and salinity

differences between coastal and offshore areas. Owing to surface cooling in winter and continuous input of river discharge, shallow coastal waters are colder and less saline than offshore oceanic waters. Due to its lower salinity, the colder coastal water does not sink, whereas oceanic saline water also cannot sink as it mixes with warmer water from the open sea (Yanagi and Isobe, 1992). Therefore, differences in water temperature and salinity between coastal and offshore areas induce a surface convergence zone, where thermohaline fronts with large horizontal temperature and salinity gradients form (Akitomo

et al., 1990; Endoh, 1977; Yanagi and Isobe, 1992). The density between the two sides of the thermohaline fronts was minimal because the salinity difference compensated for the temperature difference. However, the cabbeling effect provides the maximum density in the front area, which is an important factor in the intensity of thermohaline fronts (Endoh, 1977; Yanagi and Isobe, 1992).

Thermohaline fronts have been identified in the Iyo-nada and Kii Channel of the SIS based on hydrographic observations

(Oonishi et al., 1978; Toda, 1992; Yanagi, 1980; Yoshioka, 1988). Yanagi (1980) showed that the thermohaline front in Iyo-nada forms from late November to late February and remains stable in its horizontal position and intensity despite significant variations in surface cooling and river discharge. However, Yoshioka (1988) showed that the thermohaline front in the Kii Channel has remarkable variability every ten days, which is likely induced by the intrusion of warm water from the Kuroshio. The Hayasui Strait's strong tidal current is proposed to explain why water intrusion does not affect the thermohaline front in



Iyo-nada (Figure 1) (Yanagi and Sanuki, 1991). However, detailed variations in these fronts and their susceptibility to ambient current remain unclear. Additionally, wind-driven currents may influence the fronts because wind patterns have been suggested to strongly influence the formation of the thermohaline front in the South China Sea (Wang et al., 2020). The relaxation or intensification of the northeasterly wind during winter can affect the onshore or offshore movement of the thermohaline front (Liu et al., 2010).

In this study, we aimed to enhance our understanding of different fronts (including tidal and thermohaline fronts) in the entire SIS by analyzing a seven-year hourly SST dataset. We also examined the dynamic processes related to the intra-tidal and spring-neap tidal cycles as well as the seasonal and anomalous frontal variabilities. Section 2 describes the data and front detection methods used in the study. The major results are presented in Section 3, followed by the discussion and summary in Section 4.

## 2 Data and Method

To estimate the multiple time-scale variations of coastal fronts, both spatial and temporal resolutions of SST data must be high. The geostationary meteorological satellite Himawari-8 became operational in July 2015. This satellite was launched by the Japan Meteorological Agency (JMA) and is located at 140.7° E, with an observation area of 80° E to 160° W and 60° N to 60° S. It was loaded with an optical radiometer, the Advanced Himawari Imager which has 16 spectral bands from the

visible to infrared (IR) wavelengths. The IR bands centered at 3.9, 8.6, 10.4, 11.2, and 12.4 μm are usable for skin SST data retrieval. The Himawari-8 satellite data had a spatial resolution of 2 km and a temporal interval of 1 h, which allowed us to examine the intra-tidal variations of the fronts. A dataset of SST retrieved from 10.4, 11.2, and 8.6 μm IR bands in nearly real-time was released by the Earth Observation Research Center of the Japan Aerospace Exploration Agency (JAXA) (Himawari Monitor, 2023) and used in this study. These SST data were compared with observational data obtained from

drifting and tropical-moored buoys, and the root-mean-square difference and bias were ~0.59 and ~-0.16 K, respectively (Kurihara et al., 2016). The comparison results agree with the mean difference between the skin and bulk temperatures reported by Donlon et al. (2012). In this study, we used SST data from January 2016 to December 2022 obtained from the JAXA website (Himawari Monitor, 2023).

Further, the gradient-based algorithm developed by Belkin and O'Reilly (Belkin and O'Reilly, 2009) was used to map fronts

with high-gradient zones in the SST data. This algorithm removes data noise based on an advanced contextual feature-preserving median filter while preserving high-gradient zones and has been widely used for front detection in satellite oceanography. The algorithm consists of two steps. First, a median filter, which is a highly efficient digital filtering technique that removes isolated noise while preserving edges in the data, was applied to preprocess the SST data to remove data noise. The second step is traditional edge detection: because the fronts are characterized by enhanced gradients, an edge

detector brings out the features processed with the contextual feature-preserving filter (Belkin and O'Reilly, 2009). The gradient magnitude and direction were then calculated for each pixel. The gradient vector is calculated using the Sobel operator, which is a simple and effective method of enhancing the visibility of edges in digital images consisting of two 3 × 3 kernels:

$$G_x = [-1\ 0\ +1;\ -2\ 0\ +2;\ -1\ 0\ +1] * T\ , \tag{1}$$

$$G_y = [+1\ +2\ +1;\ 0\ 0\ 0;\ -1\ -2\ -1] * T\ , \tag{2}$$

where $T$ is the SST data processed in the first step; $G_x$ and $G_y$ are two images containing approximations for derivatives in the X (eastward) and Y (northward) directions, respectively, and * is the convolution sign (Belkin and O'Reilly, 2009). The gradient magnitude ($T_G$, ℃/km) of $G_x$ and $G_y$ was utilized to define the intensity of the fronts as follows

$$T_G = \sqrt{\left(\frac{G_x}{\delta x}\right)^2 + \left(\frac{G_y}{\delta y}\right)^2}\ , \tag{3}$$



where $x$ and $y$ represent the eastward and northward coordinators, respectively. Generally, pixels with $T_G \geq 0.1\ °C/km$ are identified as fronts. The front frequency of each pixel was calculated as follows:

$$FF = \frac{N_a}{N_b},\qquad(4)$$

where $N_a$ is the number of occurrences of fronts in a given pixel and $N_b$ is the number of cloud-free valid data in a given pixel.

Identifying the SST data phase in a tidal cycle is crucial to examine the intra-tidal variations in the front. Dong and Guo (2021) noted that the intra-tidal movement of the tidal front is mainly controlled by tidal current advection. In the SIS, the semidiurnal tide prevails across most of the sea. Tidal waves in the eastern and western regions are progressive waves that become stationary waves in the central region. The phase difference between the tide and tidal currents is approximately 90° in the central region and is much smaller in the eastern and western regions (Takeoka, 2002). Tidal current data in the SIS are limited, but many gauge stations exist from east to west. Therefore, we used the phase of the tide data to study the intra-tidal variations of the fronts in the SIS after applying the phase relationship between the tide and tidal current.

We utilized an hourly tide-level dataset from seven tide gauge stations along the SIS coast (Figure 1a), obtained from JMA, to examine the relationship between the tide phase and intra-tidal variations of fronts. Owing to the influence of clouds, SST data are not always continuous. To address this data discontinuity issue when compositing an SST dataset with the other data at the same tide phase, we divided a semidiurnal tide cycle into eight intervals and established nine phase indexes, starting at high-water tide, progressing through the ebb tide phase, reaching low-water tide, then through flood tide phase, and back to high-water tide (numbers 1–9 in Figure S1). These phase indices facilitated the averaging of SST data within the same phase. Dividing a semidiurnal tidal cycle (~12 h) into nine phases effectively tracks intra-tidal changes in tidal fronts (Dong and Guo, 2021). Given the use of real tide levels comprising various tidal constituents, the tide level at the first high water level (number 1 in Figure S1) often differs from that at the second high water level (number 9 in Figure S1).

With access to a substantial SST dataset, we comprehensively estimated the inter-tidal variations of the fronts in the SIS, including variations over the month, between spring and neap tides, and the influence of winds on these fronts.

### 3 Results

Figure 1b shows the occurrence frequency of fronts in the SIS, calculated using seven years of SST data, revealing their presence in nearly all basins. Tidal fronts in the SIS mainly form around narrow straits characterized by significant spatial variations in tidal currents, resulting in the identification of 10 tidal fronts around these straits (Table 1). In contrast, fronts in the western part of Iyo-nada and the southern mouth of the Kii Channel are thermohaline fronts (Table 1).

**Table 1.** Locations of fronts and the nearest tidal gauge and wind data stations.

| Sign | Location | Nearest gauge data station | Nearest wind data station |
|---|---|---|---|
| F1 | Between HyS and Bungo Channel | G1 | *Seto* |
| F2 | Between HyS and Iyo-nada | G1 | *Seto* |
| F3 | Between Tsurushima Strait (TrS) and Iyo-nada | G2 | *Minamiyoshida* |
| F4 | Between Tsurushima Strait (TrS) and Hiroshima Bay | G2 | *Otake* |
| F5 | Between Kurushima Strait (KrS) and Hiuchi-nada | G3 | *Imabari* |



| F6 | Between Akashi Strait (AkS) and Harima-nada | G4 | *Akashi* |
|---|---|---|---|
| F7 | Between Naruto Strait (NrS) and Harima-nada | G6 | *Akashi* |
| F8 | Between Akash Strait (AkS) and Osaka Bay | G5 | *Kobekuko* |
| F9 | Between Naruto Strait (NrS) and Kii Channel | G7 | *Nandan* |
| F10 | Between Kitan Strait (KtS) and Kii Channel | G7 | *Nandan* |
| F11 | Western part of the Iyo-nada | G1 | *Musashi* |
| F12 | Southern mouth of the Kii Channel | G7 | *Kamoda* |


Figure 2 shows the seasonal mean SST and its gradient magnitude ($T_G$) from spring to winter in the SIS. The tidal fronts of F1 to F10 (Table 1) appear in summer, and their intensity is larger than 0.1 ℃/km. The thermohaline fronts of F11 and F12 (Table 1) appear in winter and early spring with an average intensity larger than 0.1 ℃/km. The tidal fronts F7, F9, and F10 were slightly weaker and smaller. Tidal front F1, forming as early as April (Dong and Guo, 2021), extends into spring.

Conversely, tidal front F9 seems to be followed by a thermohaline front during winter. Shapes of tidal fronts in the SIS primarily align orthogonally with the direction connecting the straits and basins, coinciding with the tidal current direction. Thermohaline front F11 runs predominantly parallel to the coastline, whereas thermohaline front F12 assumes an S-shaped configuration, likely influenced by the Coriolis effect (Toda, 1992).

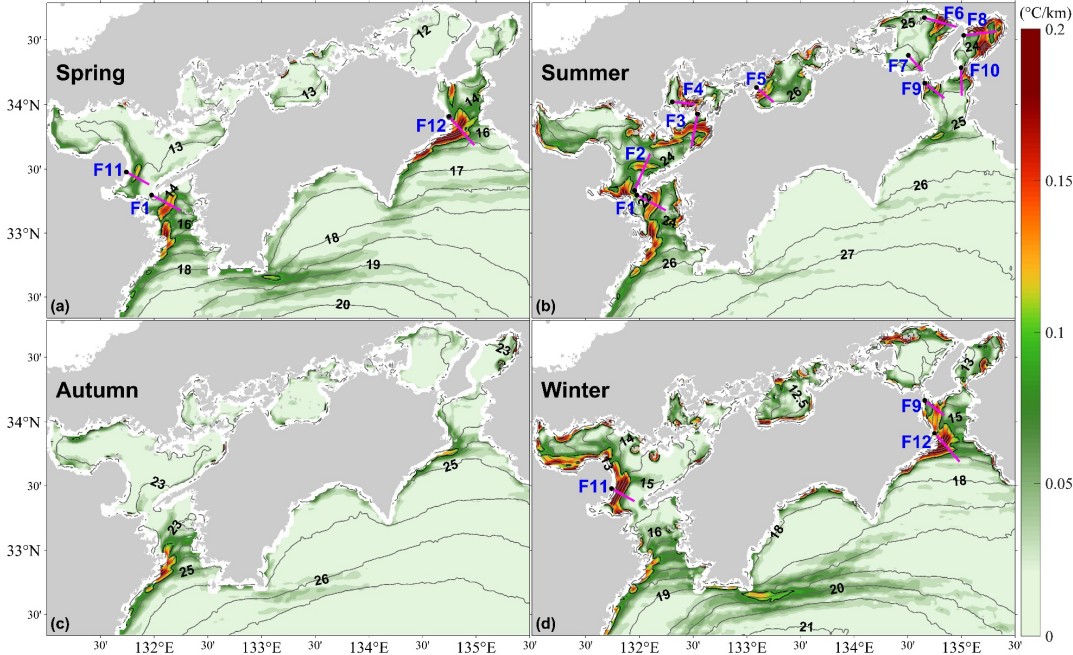

**Figure 2**. Seasonal mean SST (contours) and its gradient magnitude ($T_G$) (colors) from spring to winter. The thick black line indicates a gradient magnitude ($T_G$) contour of 0.1 ℃/km.



### 3.1 Monthly variations of the fronts

Figures 3 and 4 show the monthly mean SST and its gradient magnitude ($T_G$) from January to December. With the characteristics and variations of tidal front F1 previously described by Dong and Guo (2021), our focus shifts to the other

nine tidal fronts and two thermohaline fronts. Apart from tidal front F9, which exhibited a different pattern, the remaining tidal fronts primarily formed in May, peaked in intensity in July, weakened in August, and disappeared in September (Figures 3 and 4). Tidal front F9 reached its maximum intensity in May, gradually weakened, and disappeared in September (Figure 4). Thermohaline front F11 emerged in December, peaked in February, weakened, and disappeared in March (Figure 3). Conversely, thermohaline front F12 endured from November to May (Figure 4), displaying a wider profile with a width

(in the direction perpendicular to the front) exceeding 10 km at its peak intensity.

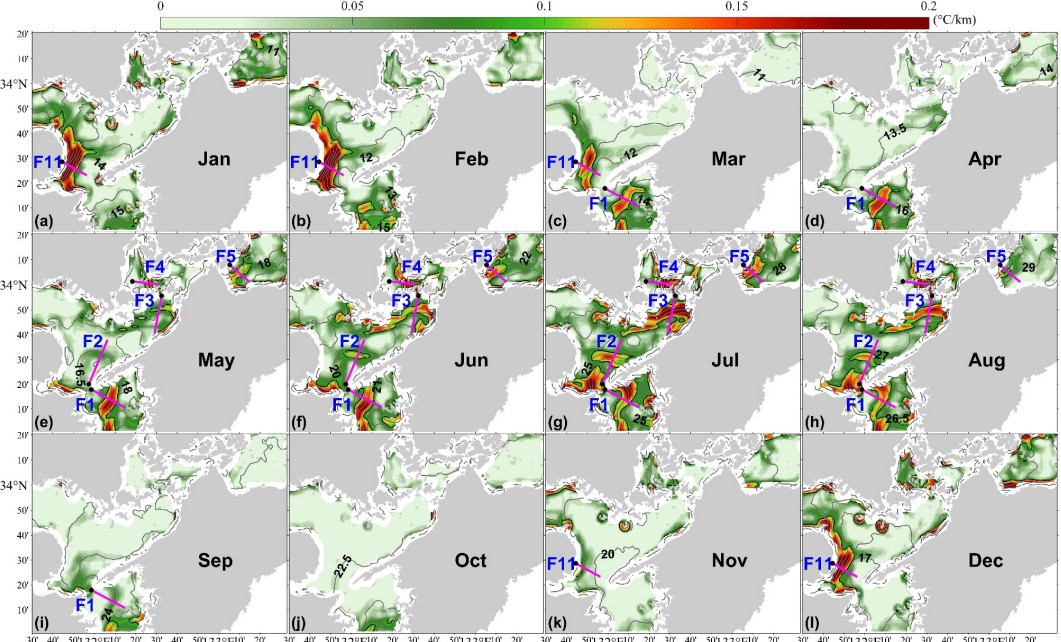

**Figure 3**. Monthly mean SST (contours) and its gradient magnitude ($T_G$) (colors) from January to December in the western part of the SIS. The pink line perpendicular to each front is used to quantify the variations of the front along it. The black dot indicates the start point of each pink line. The thick black line indicates a gradient magnitude ($T_G$) contour of 0.1 ℃/km.




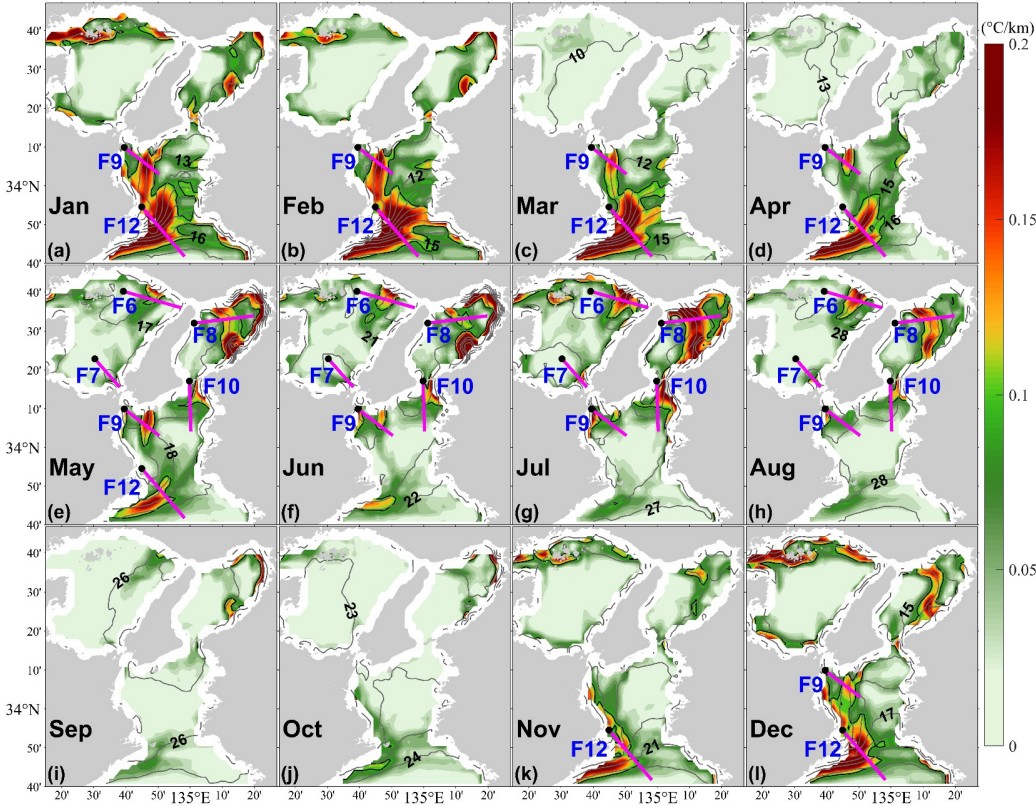

**Figure 4**. Monthly mean SST (contours) and its gradient magnitude ($T_G$) (colors) from January to December in the eastern part of the SIS. The pink line perpendicular to each front is used to quantify the variations of the front along it. The black dot indicates the start point of each pink line. The thick black line indicates a gradient magnitude ($T_G$) contour of 0.1 ℃/km.

Figure 5 shows the monthly variations in the gradient magnitude ($T_G$) of SST along the pink lines across the fronts in Figures 3 and 4. The maximum intensities of the monthly averaged fronts from F2 to F12 were approximately 0.15, 0.2, 0.25, 0.15, 0.18, 0.13, 0.2, 0.16, 0.13, 0.3, and 0.32 ℃/km respectively. Owing to the spatial variation in the intensity of the fronts, the maximum intensity of the fronts along these pink lines may not be the maximum value of the fronts (Figures 3 and 4). If the monthly averaged fronts were bounded by a gradient magnitude ($T_G$) of 0.1 ℃/km, the width of these fronts varied from 205  about 5 km to 10 km (Figure 5).



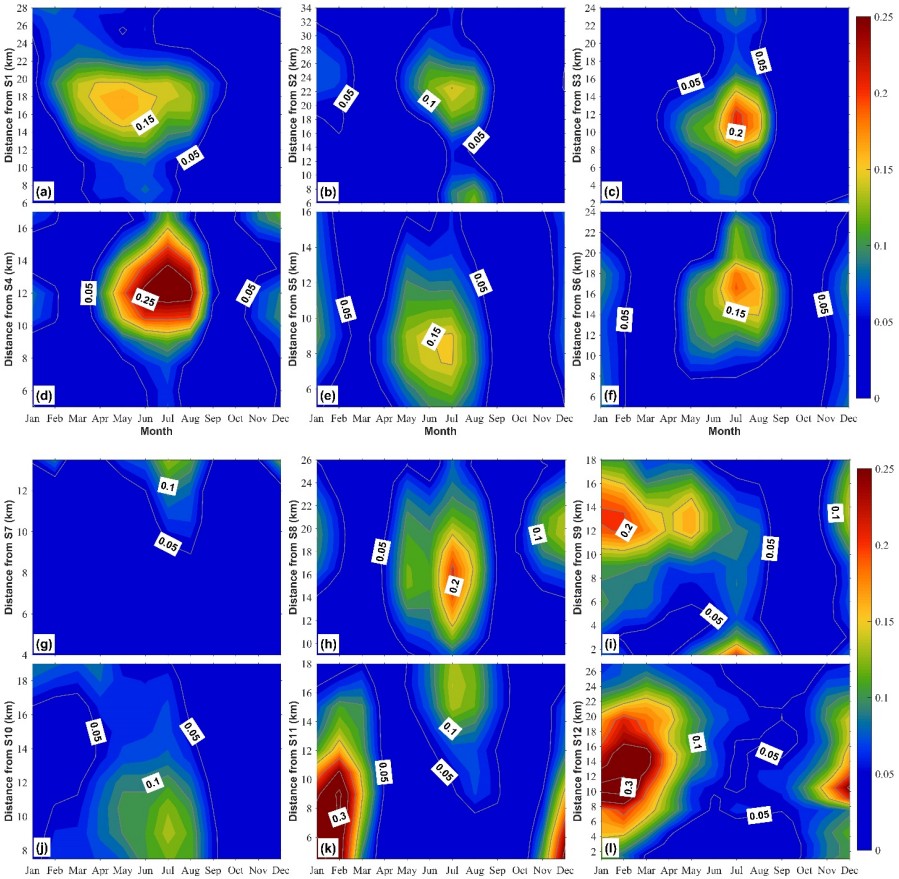

**Figure 5**. Monthly variations of the gradient magnitude ($T_G$) of the SST along the pink lines across each front in Figures 3 and 4. S1-S12 in the y-axis labels means the start point of each pink line (black dot in Figures 3 and 4).

## 3.2 Spring-neap tidal cycle variations of the fronts

To determine the spring-neap tidal cycle variations in the position and intensity of the fronts, we averaged the SST data during the neap and spring tides separately and then calculated the corresponding gradient magnitude ($T_G$) (Figure 6). First, 12 fronts were assigned to each of the 12 areas. Then, the SST data over the front area for the neap (spring) tide period were selected according to the neap (spring) tide period known from tide-level data at the nearest tidal gauge station (Table 1). For example, tide-level data at tidal gauge station G3 (Figure 1a) were used to determine the time for averaging the SST data over the area for tidal front F5 (Figure 6 and Table 1). Utilizing the tide levels at different gauge stations and the same gauge station does not influence the average result for the neap and spring tides, but it does influence the estimation of the intra-tidal variations of the fronts because of the substantial change in the tide phase over the SIS (Guo et al., 2013).



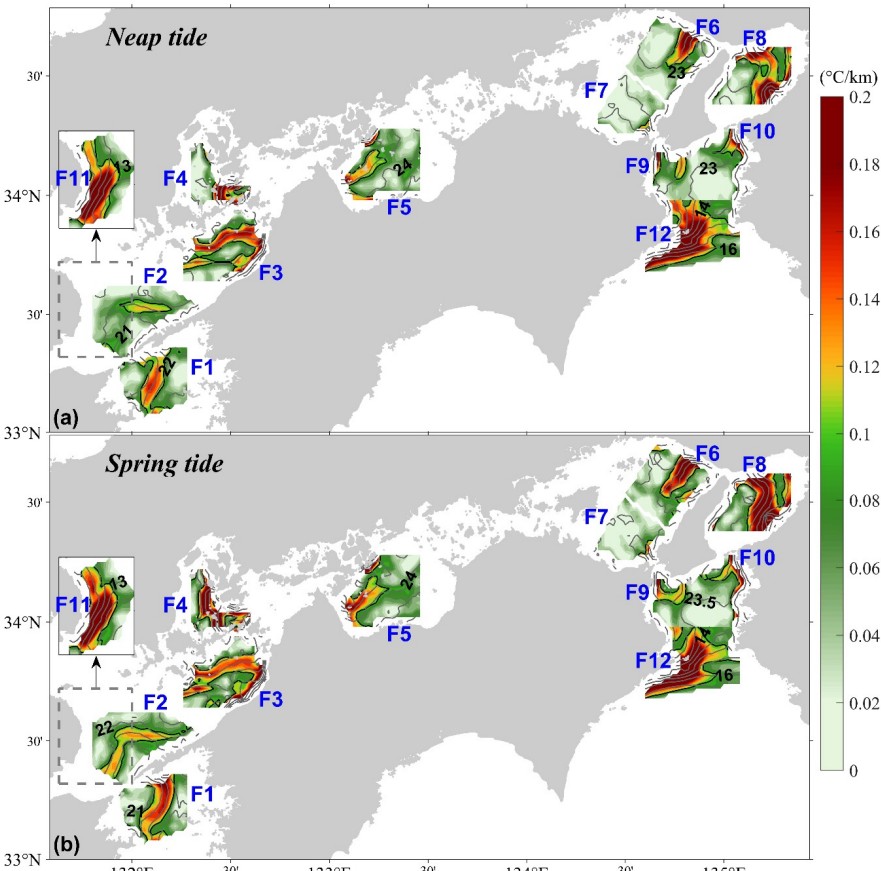

**Figure 6**. Average SST (contours) and its gradient magnitude ($T_G$) (colors) in the SIS during (a) neap tide and (b) spring tide. The thick black line indicates a gradient magnitude ($T_G$) contour of 0.1 ℃/km. The dashed box indicates the original location of the front F11 area.

While the spring-neap tidal cycle variation in the averaged position of these fronts showed minimal change (Figure 6), the intensity of all averaged fronts, including the thermohaline fronts, was generally higher during spring tide than during the neap tide (figures not shown here). However, for some fronts in certain months, the monthly averaged frontal intensity was lower during the spring tide than during the neap tide, which may be caused by the changes in background stratification and residual currents (Dong and Guo, 2021).

### 3.3 Intra-tidal variations of the fronts

In this study, a composite SST dataset consisting of data from the same tidal phase was utilized to determine intra-tidal variations in the position and intensity of the fronts. As mentioned in Section 2, we used nine tide phases in a semidiurnal tide cycle (numbers 1–9 in Figure S1) as an index to average the SST data at the same tide phase. Since the intra-tidal variations of the tidal front are more significant during the spring tide than during the neap tide and these variations are not dependent on the month (Dong and Guo, 2021), we used the SST data during the spring tide in all summer months of 2016–2022 for this composite analysis. Similarly, we used the SST data in all winter months to complete the composite analysis of the thermohaline front. The SST data for a specific phase were selected based on the tidal phase of the tide-level data at the nearest tidal gauge station (Table 1).



Figures 7 and 8 show the average SST across the areas of the 12 fronts and the gradient magnitude ($T_G$) at the nine tidal phases. For tidal front F1, during the ebb tide (Phases 1–5), the front shifted toward the stratified area, exhibiting increased intensity (Figures 7a-7d). By the low-water tide (Phase 5), the front reached its farthest point from the HyS, displaying maximum intensity (Figure 7e). Throughout the flood tide (Phases 5–9), the front moved towards the mixed area, resulting in decreased intensity (Figures 7f-7h). By the second high-water tide (Phase 9), the front did not return to its initial position from the first high-water tide (Phase 1), and its intensity weakened (Figure 7i). Since the tidal waves here are mainly progressive, the tide closely aligns with the tidal current phase (Takeoka, 2002). Consequently, the southward movement of the tidal front (Figures 7a-7e) corresponds to ebb tidal currents, and the northward movement of the front (Figures 7e-7i) corresponds to flood tidal currents. The maximum movement of the tidal front along the pink lines in Figure 7 during the ebb tidal currents or the flood tidal currents was ~5 km, and the intensity changed by 0.2 ℃/km (Figure 9a). These intra-tidal variations in the tidal front F1 are consistent with those reported by Dong and Guo (Dong and Guo, 2021). In addition, the asymmetry in the position and intensity of the front at the first and second high-water tides was likely due to the asymmetry of the ebb and flood tides.

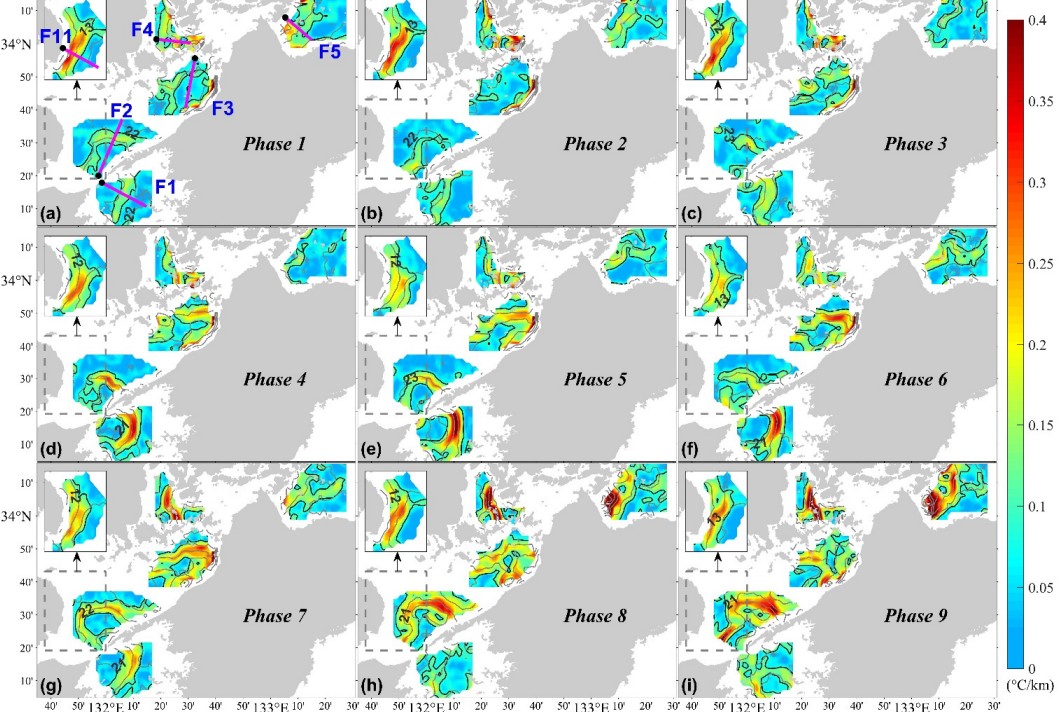

**Figure 7**. Averaged SST (contours) over the front F1-F5, F11 areas in the western part of the SIS and its gradient magnitude ($T_G$) (colors) for the nine tide phases (numbers 1-9 in Figure S1). The pink line perpendicular to each front is used to quantify the variations of the front along it. The black dot indicates the start point of each pink line. The thick black line indicates a gradient magnitude ($T_G$) contour of 0.1 ℃/km. The dashed box indicates the original location of the front F11 area.



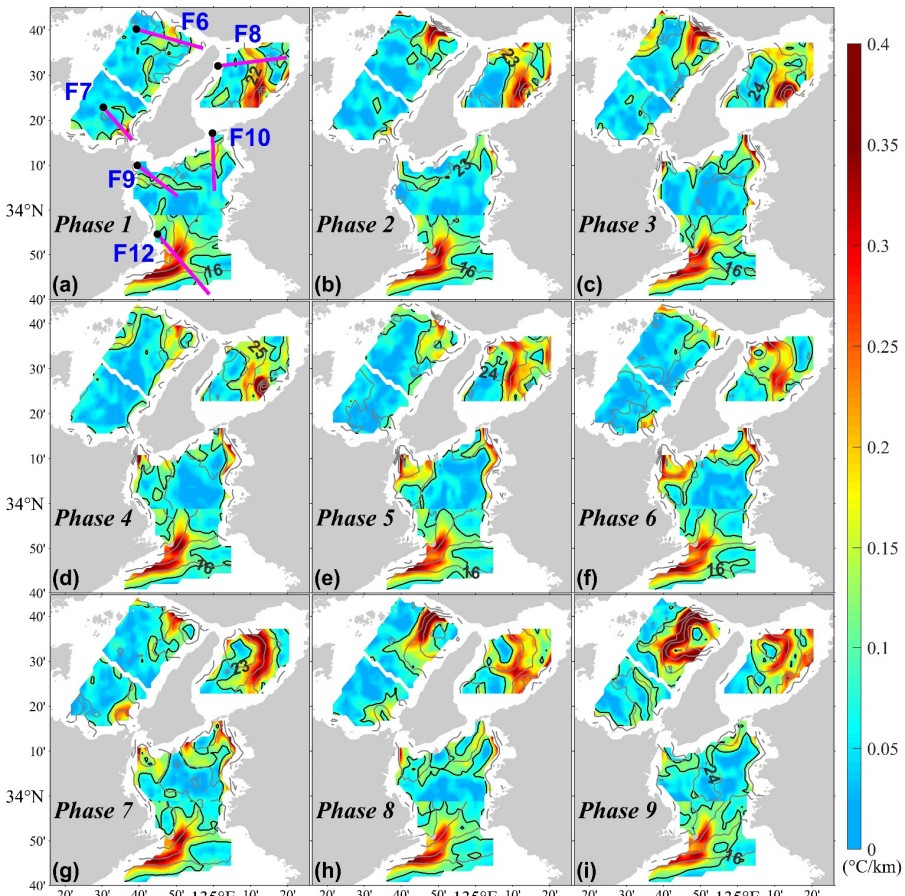

**Figure 8**. Averaged SST (contours) over the front F6-F10, F12 areas in the eastern part of the SIS and its gradient magnitude ($T_G$) (colors) for the nine tide phases (numbers 1-9 in Figure S1). The pink line perpendicular to each front is used to
quantify the variations of the front along it. The black dot indicates the start point of each pink line. The thick black line indicates a gradient magnitude ($T_G$) contour of 0.1 ℃/km.

The phase difference between the tide and tidal currents increased closer to the central region of the SIS, and the phases of the high-water (low-water) and ebb (flood) tidal currents gradually deviated. For tidal front F2, mixed water was on its south
side, and stratified water was on its north side, which is opposite to the situation of tidal front F1. Therefore, the intra-tidal variations of front F2 during ebb or flood tidal currents should be reversed from those of front F1. From Phases 1 to 5, front F2 moved approximately 5 km closer to the Hayasui Strait, and the intensity of the front changed irregularly (Figures 7a–7e, and 9b). From Phases 5 to 9, it moved nearly 9 km to the stratified area, and its intensity increased by ~0.1 ℃/km (Figures 7a–7e, and 10b). Although the intensity of front F2 did not decrease when the front moved toward the mixed area, the
intensification during moving from the mixed area to the stratified area was consistent with that of front F1. As suggested by Dong and Guo (2021), the effect of tidal straining on driving convective mixing intensifies the front because heavy water from the mixed area is brought above the light water in the stratified area.

The maximum intra-tidal movement of the tidal front F3 in one direction was ~5 km (Figures 7c-7f, and 9c) and the corresponding change in the intensity was ~0.1 ℃/km (Figure 9c). The intensification (weakening) of front F3 also occurred
during the front movement towards the stratified (mixed) area. For the tidal front F4, the intra-tidal movement of the front



was small, but the intra-tidal change in the intensity of the front was large, about 0.3 ℃/km (Figure 9d). This is likely because the amplitude of the tidal current is minimal in Hiroshima Bay but substantial in Tsurushima Strait, owing to the complex horizontal geometry around the strait (Takeoka, 2002). Therefore, the intra-tidal movement of the front is mainly controlled by tidal current advection and is small in places where the tidal current is weak. In the strait, the strong tidal

current induces large advection and strong mixing, intensifying and squeezing the front to the stratified area (Figure 9d). The characteristics of the currents around the tidal front F5 were similar to those around F4; the amplitude of the tidal current in the mixed Kurushima Strait area (Figure 1) was much larger than that in the stratified Hiuchi-nada area (Figure 1). Therefore, the intra-tidal variation in the position of the front F5 was small, whereas the variation in the intensity of this front was large (Figure 9e).

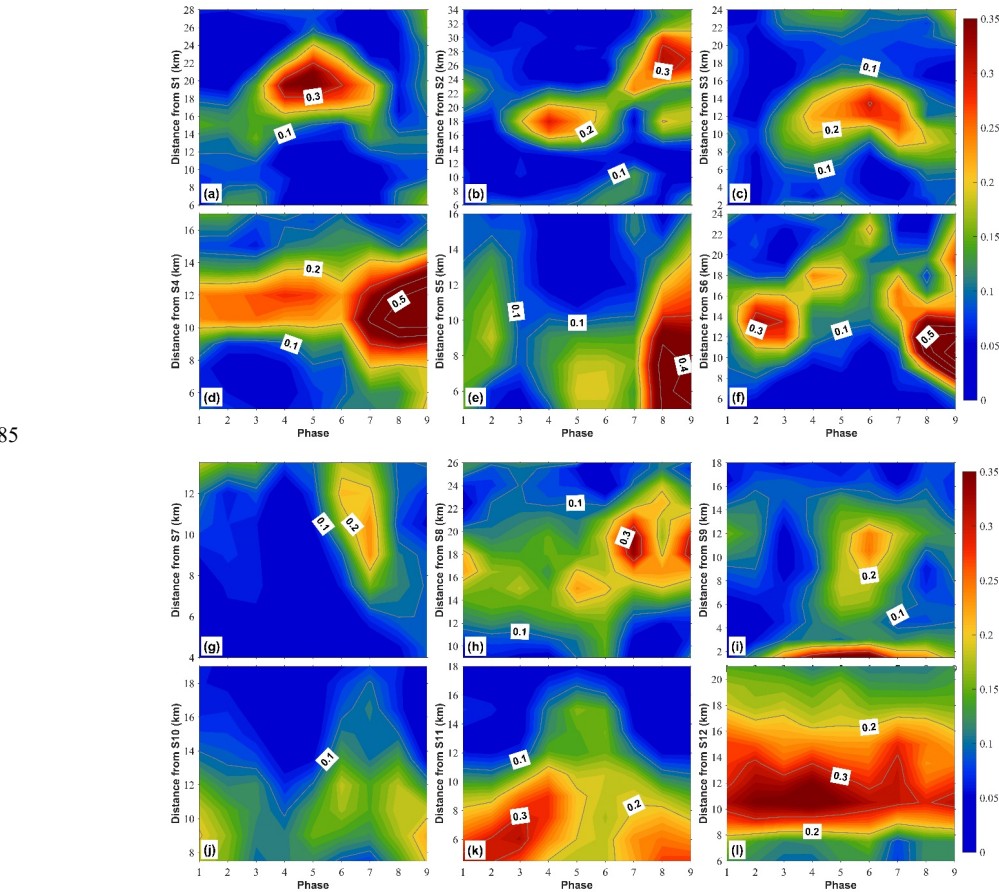


**Figure 9**. Intra-tidal variations of the gradient magnitude ($T_G$) of the SST along the pink lines across each front in Figures 7 and 8. S1-S12 in the y-axis labels means the start point of each pink line (black dot in Figures 7 and 8).

In the eastern region of the SIS, the amplitude of the $M_2$ tide is small (Guo et al., 2013), which decreases the irregularity of the tide level. The intra-tidal variations in the position and intensity of tidal front F6 were apparent. From Phases 1 to 2, the front moved a little (~3 km) to the stratified area, and its intensity increased by ~0.17 ℃/km (Figures 8a–8b, and 9f). From Phases 3 to 6, the front moved ~8 km in the opposite direction towards the mixed Akashi Strait area, and its intensity decreased by ~0.15 ℃/km (Figures 8c–8f, and 9f). From Phases 7 to 9, the front moved nearly 11 km toward the stratified

area again, and the intensity of the front increased by at least 0.4 ℃/km (Figures 8g-8i, and 9f). For the tidal front F8 at the east side of the Akashi Strait, the maximum intra-tidal movement of the front in one direction was ~4 km (Figures 8d-8g, and





9h), and the corresponding change in the intensity was ~0.15 ℃/km (Figure 9h). The intra-tidal variations in the F8 front were less substantial than those in the front F6. The direction of the tidal currents around front F8 in Osaka Bay (Figure 1) is mainly along the direction of an arc connecting the Akashi and Kitan Straits (Figure 1) (Guo et al., 2013). Consequently,

farther away from Akashi Strait, the direction of the tidal currents gradually became parallel to the front. Because tidal fronts F7, F9, and F10 are relatively weak and are therefore easily affected by other factors, such as background currents and winds, the intra-tidal variations in the position and intensity of these fronts are not apparent (Figures 8, 9g, 9i, and 9j).

The intra-tidal variations of the thermohaline front F11 in the western part of Iyo-nada were similar to the intra-tidal variations of tidal front F2 in the central part of Iyo-nada. Both fronts shifted towards the Hayasui Strait during the ebb tidal

current with decreased frontal intensity (Figures 7 and 9k). However, the range of intra-tidal variations of the thermohaline front F11 was smaller than that of tidal front F2. The direction of the tidal current is perpendicular to front F2 but not to front F11 (Guo et al., 2013). Intra-tidal variations in the thermohaline front F12 were minimal, likely due to the very weak tidal current and its non-perpendicular alignment with this S-shaped front (Figures 8 and 9l).

### 3.4 Impacts of wind on the fronts

In this study, we utilized wind data collected by the Automated Meteorological Data Acquisition System, managed by the JMA at nine observation stations (Figure 10) to examine the influence of wind on variations in the fronts. The temporal resolution of these wind data was also one hour, including wind speed and direction. Figure 10 shows a succinct view of how the speed and direction of the wind in summer from 2016 to 2022 are typically distributed at *Seto, Minamiyoshida, Otake, Imabari, Akashi, Kobekuko*, and *Nandan* stations and in winter at *Musashi* and *Kamoda* stations. Using polar gridding

coordinates and wind direction, we plotted the frequency of winds over the period, with color bands showing the wind speed range. Eight directions of wind distribution were present, and the direction of the longest spoke indicated the wind direction with the greatest frequency. For example, the wind at station *Seto* is mainly south wind in the summer from 2016 to 2022, accounting for about 55% of the eight wind directions, and the maximum speed of the south wind is ~12 m/s (Figure 10) for the wind at station *Minamiyoshida*, the proportion of each direction of the wind is close, and the maximum wind speed in all

directions is ~8 m/s. The wind in winter near the *Musashi* and *Kamoda* stations is mainly westerly.

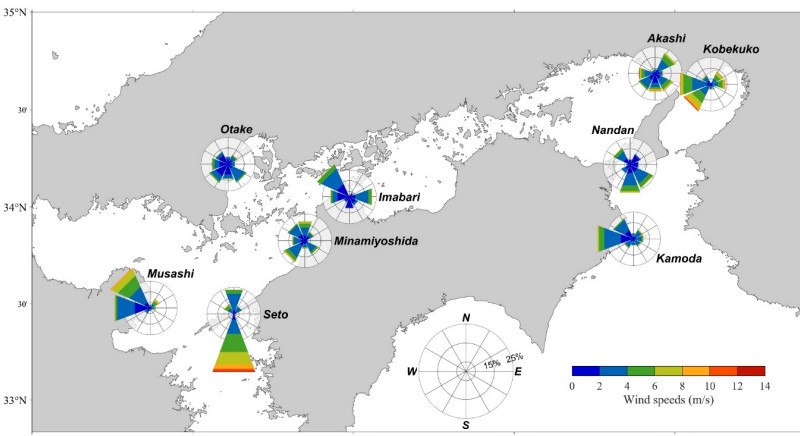

**Figure 10**. A succinct view of how the speed and direction of the wind data from 2016 to 2022 are typically distributed at nine observation stations. The wind data at *Seto, Minamiyoshida, Otake, Imabari, Akashi, Kobekuko*, and *Nandan* are for the summer, while the wind data at *Musashi* and *Kamoda* are for the winter. The frequency of winds is plotted by wind direction

using a polar coordinate of gridding, and color bands show the wind speed ranges. The distribution of the wind is divided into eight directions.

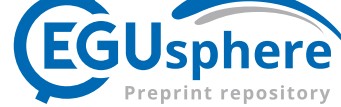

Figures 11 and 12 show the average SST corresponding to the wind data in the same direction over the 12 front areas and the gradient magnitude ($T_G$) for the eight wind directions. Wind data corresponding to each front area were obtained from the

closest observation station (Table 1). The wind over the areas of tidal fronts F1 and F2 was mainly in the north-south direction. The position of front F1 changed minimally during the south (Figure 11a) and north wind period (Figure 11e), but showed a slight intensification from the south to the north wind period. From the south to the north wind period, the variations in front F2 were opposite to those in front F1, and front F2 moved slightly closer to the Hayasui Strait with a decrease in intensity (Figures 11a and 11e). The wind at *Minamiyoshida* close to the front F3 area was nearly evenly

distributed in all directions, and the wind speed in each direction was almost the same, with an average speed of 2.8 m/s. During periods of different wind directions, there was no change in the position of front F3, but the intensity of the front decreased during periods of eastern, southeastern, and southern winds (Figure 11). The wind at *Otake* was weaker than that at *Minamiyoshida*, with an average speed of 2.1 m/s, and had limited influence on the position or intensity of front F4 (Figure 11). For front F5, although the wind at *Imabari* near the front area was also relatively weak, with an average speed of

2.1 m/s, the intensity of the front in different wind directions was different. The intensity of the front was weaker during the clockwise periods from the northeast wind to the south wind (Figures 11f-11h, 11a) than during the clockwise periods from the southwest wind to the north wind (Figures 11b-11e).

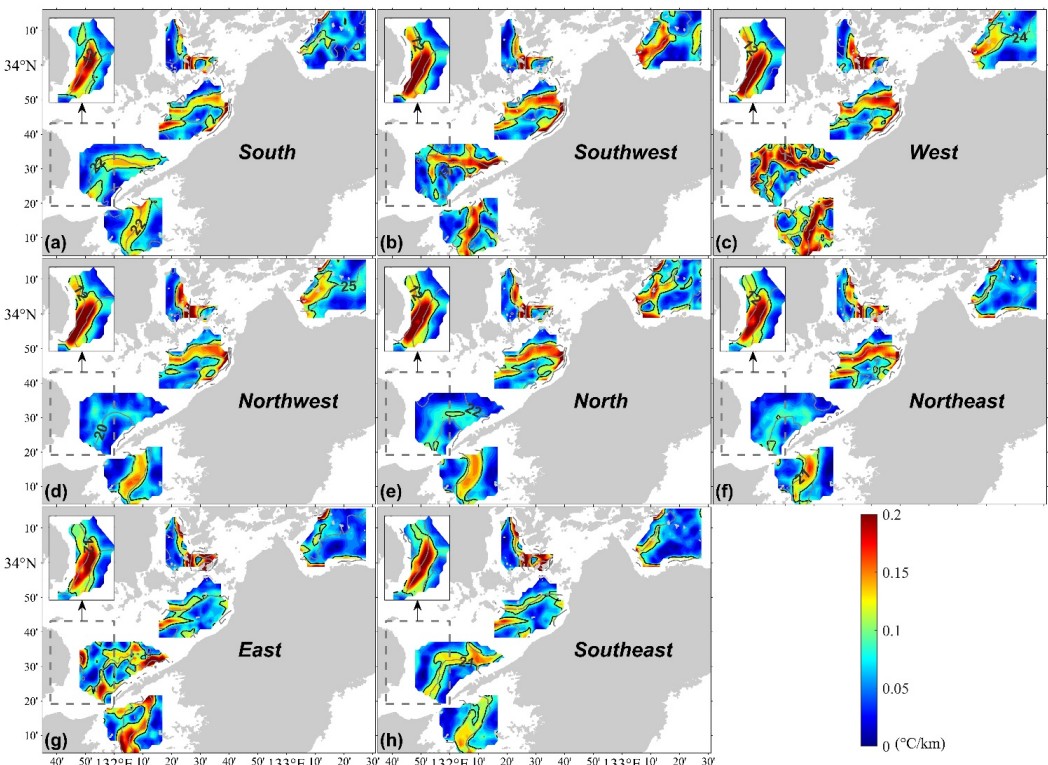

**Figure 11**. Average SST (contours) corresponding to the wind data in the same direction over the fronts F1-F5, F11 areas in

the western part of the SIS and its gradient magnitude ($T_G$) (colors) for eight directions of the wind. The thick black line indicates a gradient magnitude ($T_G$) contour of 0.1 ℃/km. The dashed box indicates the original location of the front F11 area.





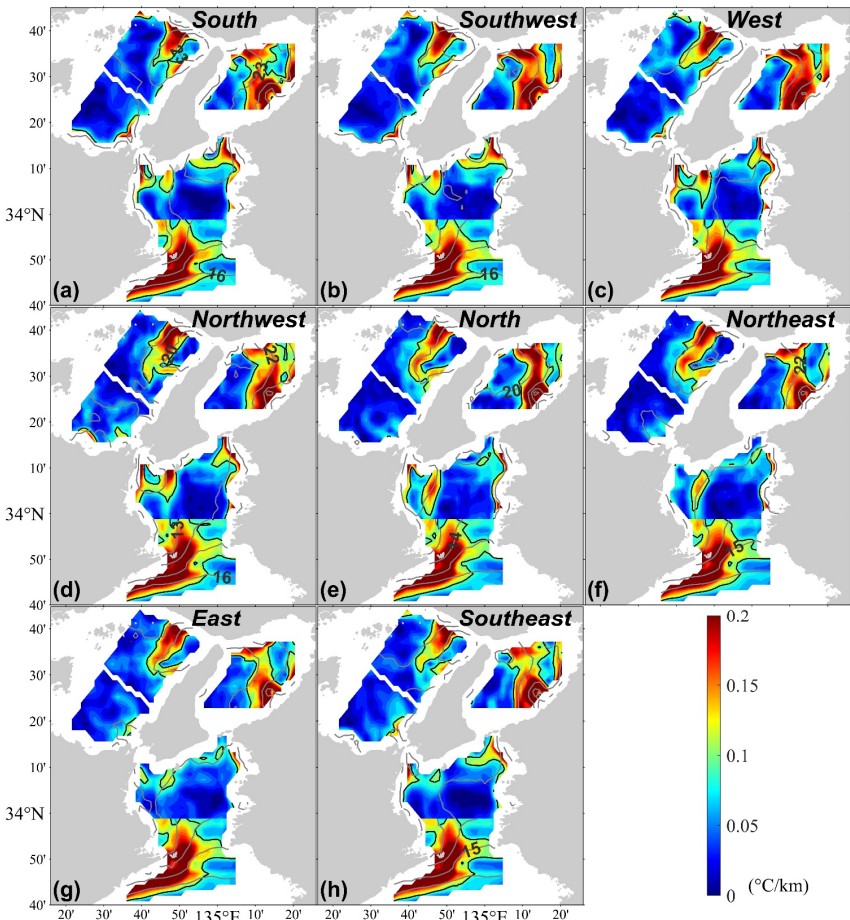

**Figure 12**. Average SST (contours) corresponding to the wind data in the same direction over the fronts F6-F10, F12 areas
in the eastern part of the SIS and its gradient magnitude ($T_G$) (colors) for eight directions of the wind. The thick black line
indicates a gradient magnitude ($T_G$) contour of 0.1 ℃/km.

The wind direction frequency distributions at *Akashi* and *Kobekuko* were similar, with average wind speeds of 3.1 and 4.0
m/s, respectively. Fronts F5 and F6, positioned on either side of the Akashi Strait, were situated farther from the strait during
the east, southeast, and south wind periods compared to other periods. Additionally, their intensities were lower during these
wind periods than those in other periods (Figure 12). This feature contrasts with the previously mentioned increase in front
intensity as the front moves toward the stratified area. The division of fronts F7, F9, and F10 in different wind directions was
based on wind data at *Nandan*, and the average wind speed was 2.3 m/s. The responses of fronts F7 and F9 on the north and
south sides of the Naruto Strait, respectively, were similar to those of fronts F2 and F1. During the southern winds, the
intensity of front F7 increased, and that of front F9 decreased, whereas that of front F7 decreased and that of front F9
increased (Figure 12). During the periods clockwise from the south wind to the north wind, front F10 gradually weakened
and then intensified during the periods from the northeast to southeast winds through the east winds. The wind in winter had
little impact on the position of thermohaline front F11 but affected its intensity, which was slightly stronger during the west
wind period, probably because the west wind squeezed the cold coastal water to Iyo-nada and then strengthened the front
(Figure 11). For the strong thermohaline front F12, the wind also had little effect on the position and intensity (Figure 12),
indicating that the thermohaline front was mainly affected by warm water intrusion from the Pacific Ocean (Toda, 1992).



**4 Discussion and Conclusion**

We used seven-year hourly SST data to examine the inter- and intra-tidal characteristics of the fronts, including tidal fronts and thermohaline fronts in the SIS. A gradient-based algorithm with an advanced contextual feature-preserving median filter

was employed to determine the position and intensity of fronts and quantify their variations. Across the SIS, ten tidal fronts exist around the narrow straits, while two thermohaline fronts are located in the western part of Iyo-nada and the southern mouth of the Kii Channel. Tidal fronts, perpendicular to the direction connecting the straits and basins, typically formed in April or May, peaked in intensity during June and July, weakened by August, and disappeared by September. Conversely, thermohaline fronts emerged in November and persisted until March or April, reaching peak intensity in February. Monthly

average positions of both types of fronts remained relatively stable. Frontal variations associated with tidal currents encompassed the spring-neap tidal cycle and intra-tidal changes. Although the positions of the fronts showed little difference between spring and neap tides, the frontal intensity was slightly larger during spring tide than neap tide.

We used a comprehensive SST dataset covering the same tidal phase within a semidiurnal tidal cycle to examine intra-tidal variations in the position and intensity of fronts. Five tidal fronts (F1, F2, F3, F6, and F8) near the Hayasui, Tsurushima, and

Akashi Straits showed significant intra-tidal variations. These fronts demonstrated increased (decreased) intensity as they shifted to stratified (mixed) water areas. The intra-tidal movement of the fronts is mainly controlled by tidal current advection. According to the effects of horizontal motion of water which has a horizontal buoyancy gradient on frontogenesis (Simpson & Linden, 1989), Dong and Guo (2021) explained that convergence or divergence of the current velocity in the direction across the front can strengthen or weaken the front (Equation 5).

$$\frac{\partial^2 T}{\partial t \partial x} = -\frac{\partial}{\partial x}\left(u\frac{\partial T}{\partial x}\right) = -\left(u\frac{\partial^2 T}{\partial x^2} + \frac{\partial u}{\partial x}\frac{\partial T}{\partial x}\right), \tag{5}$$

where $x$ is the distance from the mixed water area to the stratified water area in the direction across the front, $u$ is the current component along $x$, $-u\frac{\partial^2 T}{\partial x^2}$ is the advection of an existing gradient, and $-\frac{\partial u}{\partial x}\frac{\partial T}{\partial x}$ is the intensification or weakening of the local gradient due to the convergence or divergence of the current in the direction across the front. Typically, $\frac{\partial^2 T}{\partial x^2}$ is approximately zero since the front was defined with a maximum temperature gradient. Additionally, the tidal current diminishes with

increasing distance from the mixed water to the stratified water, leading to $\frac{\partial u}{\partial x} < 0$ when the tidal current flows from the mixed to the stratified water area, indicating convergence. Since SST is larger in the stratified area than in the mixed area, $\frac{\partial T}{\partial x} > 0$, and consequently $\frac{\partial}{\partial t}\left(\frac{\partial T}{\partial x}\right) > 0$, indicating the intensity of the front increases when the front moves to the stratified water area.

In this study, we utilized a two-dimensional depth-integrated hydrodynamic tide model covering the entire SIS with a spatial

resolution of 1 km (Guo et al., 2013) to estimate the tidal currents induced by intra-tidal variations of the tidal fronts. Our results reasonably reproduced the amplitudes and phases of the $M_2$, $S_2$, $K_1$, and $O_1$ tides and tidal currents (Guo et al., 2013). The distributions of $\log_{10}(h/u^3)$ (where $u$ is the amplitude of the $M_2$ tidal current and $h$ is the water depth) align closely with the Simpson and Hunter parameters $(h/u^3)$ (Simpson and Hunter, 1974), particularly in the regions where tidal fronts are observed in the SIS. Specifically, these areas correspond to those where the critical value of $\log_{10}(h/u^3)$ exceeds 2.5 but is

less than 3 (Yanagi and Okada, 1993) (Figure S2). Figures 13 and 14 show the average tidal currents (arrows) corresponding to the nine phases in one semidiurnal tidal cycle obtained using the composite method described in Section 3.3. The intra-tidal movement of the tidal fronts corresponds to ebb and flood tidal currents. For example, during the ebb tidal current phase, tidal front F1 moved towards the stratified water area (Figures 13b-13e), whereas during the flood tidal current phase, this front moved towards the mixed water area (Figures 13f-13i).



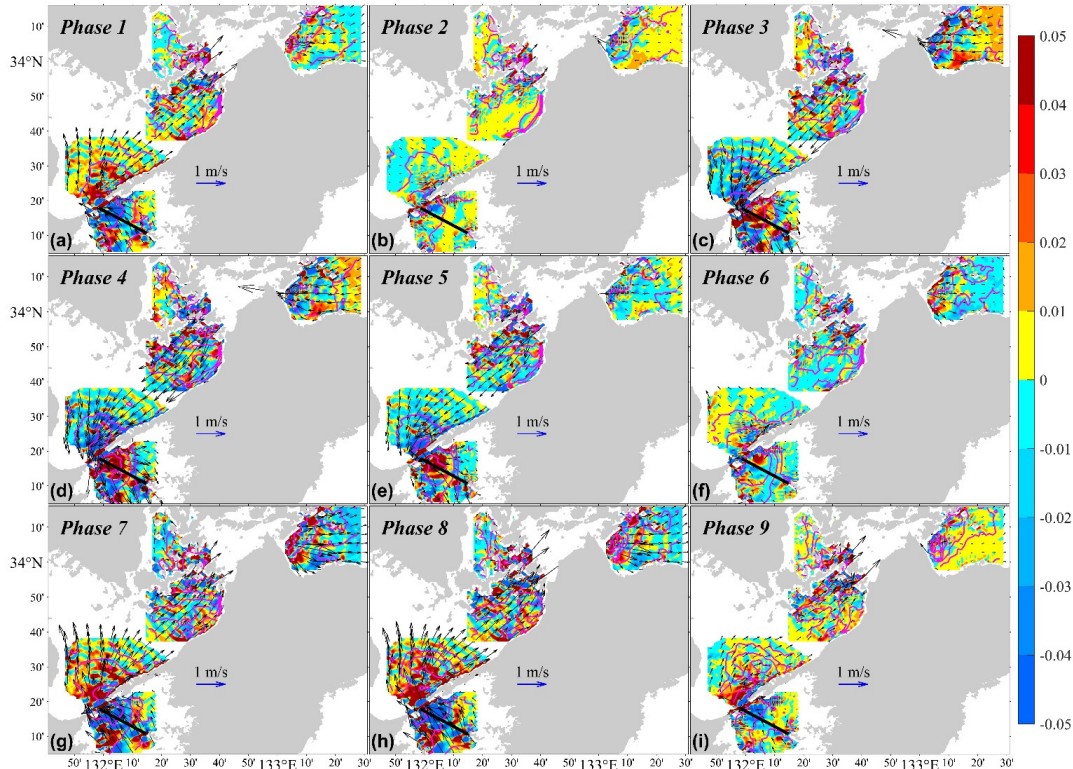


**Figure 13**. Averaged SST (contours), the corresponding tidal currents (arrows), and the divergence of the tidal currents (colors, units: $10^{-3}$ s$^{-1}$) over the tidal fronts F1-F5 areas in the western part of the SIS for the nine tide phases (numbers 1–9 in Figure S1). The thick black line perpendicular to the tidal front F1 is used to quantify the relationship between the variations of the front and the tidal currents along this direction.

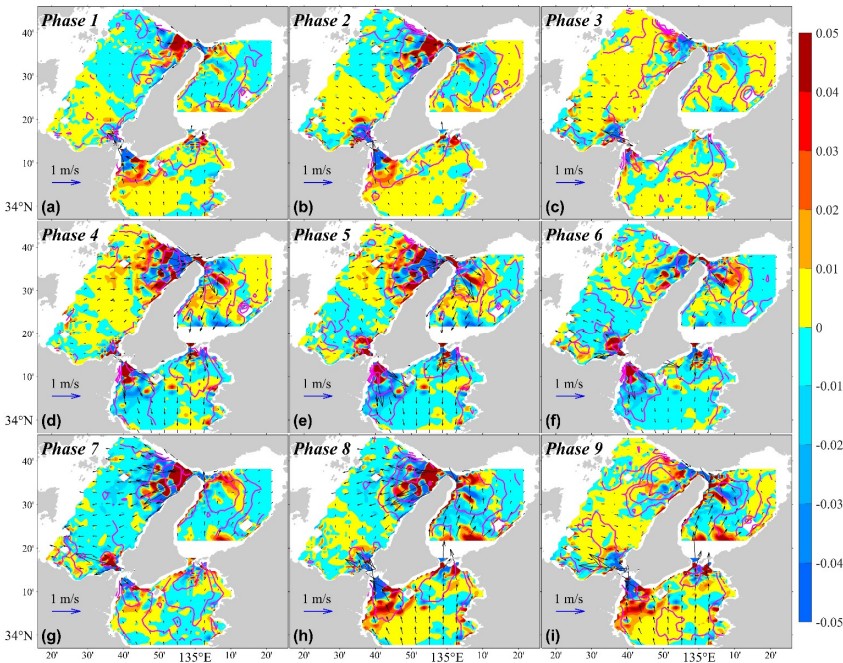






**Figure 14**. Averaged SST (contours), the corresponding tidal currents (arrows), and the divergence of the tidal currents (colors, units: $10^{-3}$ s$^{-1}$) over the tidal fronts F6–F10 areas in the eastern part of the SIS for the nine tide phases (numbers 1–9 in Figure S1).

To investigate the effect of the tidal currents on the intra-tidal variations in the frontal intensities, we separated each of the terms in Equation 5 along a line across front F1 in Figure 13 by combining the SST data with the model tidal currents (Figure 15). During the ebb tidal currents, the front's intensity increased most from Phases 3 to 4 (red line in Figure 15c), and at the front central point $-u\frac{\partial^2 T}{\partial x^2}$ was small and close to zero (pink line in Figure 15c) and $-\frac{\partial u}{\partial x}\frac{\partial T}{\partial x}>0$ (blue line in Figure 15c), indicating an increase in the intensity. Therefore, the front was intensified by the convergence of the tidal current.

Conversely, during flood tidal currents, the front moved to the mixed water area and was stretched by the divergence of the tidal currents, resulting in a decrease in its intensity (Figure 15g).

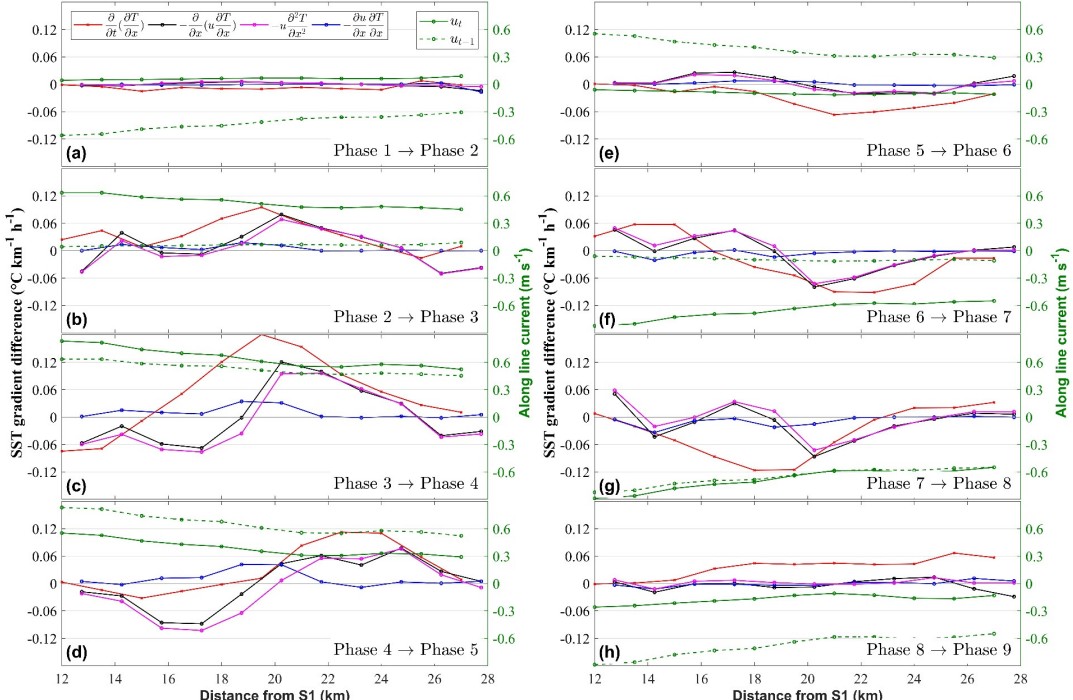

**Figure 15**. The left axis corresponds to the difference in front intensity between the two phases (red line), the transition term in Equation 5 ($-\frac{\partial}{\partial x}\left(u\frac{\partial T}{\partial x}\right)$, black line), advection of an existing gradient ($-u\frac{\partial^2 T}{\partial x^2}$, pink line), and intensification or weakening

of the local gradient due to the convergence or divergence of the current ($-\frac{\partial u}{\partial x}\frac{\partial T}{\partial x}$, blue line) along the thick black line in Figure 13. The right axis corresponds to the tidal current along the thick black line in Figure 13 at the current phase (green solid line) and the previous phase (green dashed line).

Among the 12 fronts, some tidal fronts exhibited significant intra-tidal variations; however, other tidal and thermohaline

fronts did not show clear intra-tidal variations. This discrepancy may stem from weak tidal current amplitudes around these fronts or the tidal current's direction not being perpendicular to the fronts, resulting in minimal advection effects. Additionally, wind plays a significant role in influencing SIS fronts, particularly in altering frontal intensity during different wind direction periods.



This study offers a comprehensive examination of inter-tidal, intra-tidal, and anomalous variations of fronts across the entire
SIS, utilizing seven years of hourly SST data. However, further investigations are warranted to elucidate dynamic processes,
particularly intra-tidal variations in the vertical direction. Future research may necessitate field surveys and high-resolution
hydrodynamic models to clarify the three-dimensional structure and variations in these fronts.

**Data Availability**

The Sea Surface Temperature (SST) data from January 2016 to December 2022 were obtained from the Japan Aerospace
Exploration Agency website (https://www.eorc.jaxa.jp/ptree/index.html; accessed on December 1, 2023).
The wind data were obtained from the Japan Meteorological Agency website
(https://www.jma.go.jp/jma/kishou/know/amedas/kaisetsu.html; accessed on December 1, 2023).
The tide gauge data were also obtained from the Japan Meteorological Agency website
(https://www.data.jma.go.jp/kaiyou/db/tide/suisan/index.php; accessed on December 1, 2023).

**Competing interests**

The authors declare that they have no conflict of interest.

**Funding**

This work was supported by the Moonshot Research and Development Program (JPNP18016), the New Energy and
Industrial Technology Development Organization (NEDO); KAKENHI Grants-in-Aid for Scientific Research (23K25944),
the Japan Society for the Promotion of Science. M. Dong was supported by the Ministry of Education, Culture, Sports,
Science and Technology, Japan (MEXT) to a project on Joint Usage/Research Center–Leading Academia in Marine and
Environment Pollution Research (LaMer).

**Acknowledgments**

The authors thank the Japan Aerospace Exploration Agency for providing SST data, and thank the Japan Meteorological
Agency for providing tide gauge and wind data.

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
