# Peer review of "Multiple time-scale variations of fronts in the Seto Inland Sea, Japan"

_EGUsphere, 2024_

## Referee Comment (RC1)

**Summary and recommendation:** This paper is an important contribution and should be published pending a major revision.

**Major concerns:**

(1) Satellite data alone are not sufficient to reveal dynamical mechanisms that maintain various fronts except for some obvious cases such as, e.g., river plume fronts. Therefore, it is not clear how the authors can tell a tidal mixing front (TMF) from a "thermohaline" front (using the authors' terminology). Perhaps, the authors relied on numerous studies by Japanese oceanographers who studied the Seto Inland Sea (SIS) from in situ data. If this is the case, then (A) such in situ studies should be cited with regard to each of 12 fronts identified in this study; (B) vertical sections of T and S across these 12 fronts should be provided in the main text or in the Supplementary Materials.

(2) The authors write about fronts "around" various straits. This is confusing. The authors should re-write such sentences, avoiding the ambiguous "around" descriptor.

(3) TMFs are typically aligned with certain isobaths. The depth of such isobaths marks the maximum depth of wintertime convective mixing. The authors apparently ignored this fundamental relation between TMFs' locations and bathymetry.

(4) Many references are incomplete. Make sure that volume number, issue number, article number, and pages are always provided when available.

(5) L140: "Generally, pixels with grad $T$>0.1 ℃/km are identified as fronts." – This threshold (used by many authors) is arbitrary. A discussion of front definitions is warranted.

(6) L175: "Shapes of tidal fronts in the SIS primarily align orthogonally with the direction connecting the straits and basins, coinciding with the tidal current direction." – See Comment #3 above.

**Comments on Figures:**

(1) Figure 1. The color scale in **Figure 1a** is not good. Use standard color scales like "jet" or "nipy_spectral" in Matlab. The color scale in **Figure 1b** is awful. Use jet or nipy_spectral.

(2) Figures 2, 3, 4, and 6: Color scales are poor. Use jet or nipy_spectral.

**Minor comments:**

L15: "spatial amplitude" (?)
L30: "crucial in" (?)
L47: "appearance frequency" (?)
L72: "intensigied" – intensified [Use spellchecker!]
L75: "intra- and month-dependent variations" (?)
L145: "...the SST data phase in a tidal cycle..." (?)
L271: "... heavy water ... above the light water…" (?) – "Dense water" would be better.
L382: "According to the effects of horizontal motion of water which has a horizontal buoyancy gradient on frontogenesis" – Re-write.

Best regards,
Igor Belkin, 2024-06-27

---

## Author Comment (AC1)

***Reply to Referee #1:***

*Summary and recommendation: This paper is an important contribution and should be published pending a major revision.*

Thanks very much for your helpful comments and suggestions. Our responses to the comments are as follows. The Referee's comments are cited in italics.

*Major concerns:*

*Satellite data alone are not sufficient to reveal dynamical mechanisms that maintain various fronts except for some obvious cases such as, e.g., river plume fronts. Therefore, it is not clear how the authors can tell a tidal mixing front (TMF) from a "thermohaline" front (using the authors' terminology). Perhaps, the authors relied on numerous studies by Japanese oceanographers who studied the Seto Inland Sea (SIS) from in situ data. If this is the case, then (A) such in situ studies should be cited with regard to each of 12 fronts identified in this study; (B) vertical sections of T and S across these 12 fronts should be provided in the main text or in the Supplementary Materials.*

Thank you for raising this issue. Theoretically, a tidal front forms in warming seasons while a thermohaline front in cooling seasons. This helps us to identify them from satellite data. Among the fronts in the Seto Inland Sea, the tidal fronts in Bungo Channel, Iyo-nada, Hiuchi-nada, and Osaka Bay (Figure 1) and the two thermohaline fronts in Iyo-nada and Kii Channel have been studied by field observations. We can therefore add the vertical distributions of temperature and salinity across these fronts to the revised Supplementary Materials (Figures S1 to S3). For the other 6 tidal fronts, we did not find in situ data for them. However, these fronts are located between the well-mixed strait area and the stratified basin area, and the value of $\log_{10}(h/u^3)$ at these fronts is between 2.5 and 3 (Figure S5). Furthermore, they appear in the warming seasons. All of these factors allow us to judge them as tidal fronts.

*The authors write about fronts "around" various straits. This is confusing. The authors should re-write such sentences, avoiding the ambiguous "around" descriptor.*

Thanks for your suggestion. We rewrote them in the revised manuscript.

*TMFs are typically aligned with certain isobaths. The depth of such isobaths marks the maximum depth of wintertime convective mixing. The authors apparently ignored this fundamental relation between TMFs' locations and bathymetry.*

According to Simpson and Hunter's energetics (Simpson and Hunter, 1974), which derived from a balance between the potential energy increase due to surface heating and turbulent kinetic energy dissipation induced by tidal stirring, the tidal front position depends on a critical value of $\log_{10}(h/u^3)$. Here $h$ is the water depth, and $u$ is the tidal current amplitude. Except for the tidal

fronts on both sides of Hayasui Strait (Figure 1), where the bathymetry is about 70 m, the bathymetry at other fronts is between 30 and 50 m. However, the critical values of $\log_{10}(h/u^3)$ at all tidal fronts are between 2.5 and 3, which is consistent with a previous study (Takeoka, 2002). Therefore, the tidal front position is determined by the water depth and the tidal current amplitude.

Because tidal fronts depend on the formation of stratification, they appear only in warming seasons. Therefore, the dependence of winter convective mixing on bathymetry should be the mechanism for different types of fronts.

*Many references are incomplete. Make sure that volume number, issue number, article number, and pages are always provided when available.*

Thank you for pointing this out. We have revised the reference.

*L140: "Generally, pixels with grad T >0.1 ℃/km are identified as fronts." – This threshold (used by many authors) is arbitrary. A discussion of front definitions is warranted.*

Thank you for raising this issue. For the tidal front with significant intra-tidal variations in Bungo Channel, the minimum front intensity is approximately 0.1 ℃/km when the front is closest to the mixed water area, and then the intensity can increase by 0.25 ℃/km as it moves toward the stratified water area (Dong and Guo, 2021). Similarly, the observed minimum intensity of the tidal front in the Iyo-nada is also slightly greater than 0.1 ℃/km (Sun and Isobe, 2006). Therefore, we define the pixels with $T_G \geq 0.1$ ℃/km as the fronts in the Seto Inland Sea. We have added related text on lines 141-145 of the revised manuscript.

*L175: "Shapes of tidal fronts in the SIS primarily align orthogonally with the direction connecting the straits and basins, coinciding with the tidal current direction." – See Comment #3 above.*

As mentioned above, the tidal front position is determined by the parameter depending on the ratio of water depth to cubic tidal current amplitude. The water depth increases gradually from the basin area to the strait area. The tidal current amplitude also increases from the wide basin area to the narrow strait area because of the decreasing of the sectional area the tidal current passes from the wide basin area to the narrow strait area. Consequently, the critical value of the parameter, i.e., the position of the front, is generally aligned with certain isobaths.

*Comments on Figures:*

*Figure 1. The color scale in Figure 1a is not good. Use standard color scales like "jet" or "nipy_spectral" in Matlab. The color scale in Figure 1b is awful. Use jet or nipy_spectral.*

*Figures 2, 3, 4, and 6: Color scales are poor. Use jet or nipy_spectral.*

Thanks for your suggestions. In revision, we have changed the color scale used in these figures.

*Minor comments:*

*L15: "spatial amplitude" (?)*

This is our mistake. What we wanted to present here is the spatial variation of tidal current amplitude. We have revised this sentence on line 15 of the revised manuscript.

*L30: "crucial in" (?)*

We have revised this word to "promote" on line 30 of the revised manuscript.

*L47: "appearance frequency" (?)*

This means the front frequency calculated by Equation 4.

*L72: "intensigied" – intensified [Use spellchecker!]*

Thanks. We have made this change in the revised manuscript and checked the manuscript again.

*L75: "intra- and month-dependent variations" (?)*

The month-dependent variation means that the fortnightly variations of the front are dependent on the month. We have revised this expression to "intra-tidal and month-dependent fortnightly variations" on line 77 of the revised manuscript.

*L145: "...the SST data phase in a tidal cycle..." (?)*

This means the SST data corresponding to different tidal phase in a tidal cycle.

*L271: "... heavy water ... above the light water..." (?) – "Dense water" would be better.*

Thanks. We have made this change in the revised manuscript.

*L382: "According to the effects of horizontal motion of water which has a horizontal buoyancy gradient on frontogenesis" – Re-write.*

Thanks. We have revised this sentence to "According to the effect of the horizontal motion of water with a horizontal buoyancy gradient on frontogenesis" on line 389 of the revised manuscript.

---

## Author Comment (AC2)

***Reply to Referee #2:***

*The manuscript is of significant importance, and I recommend its publication, but it requires major revisions.*

Thanks very much for your helpful comments and suggestions. Our responses to the comments are as follows. The Referee's comments are cited in italics.

*Figure 1a: The colorbar displays the full range of colors, but these colors are unusual. Using pyGMT or an equivalent tool in MATLAB might offer better and more common maps.*

*Figure 1b: Please use a linear gradient colormap.*

Thanks for your suggestions. In the revision, we have changed the color scale for the figure.

*You cite many articles, but you don't summarize their findings. For example, lines 50-53 show a large amount of papers that are not explained.*

Thank you for raising this issue. Most of these studies on the fronts in the Seto Inland Sea mainly focused on the formation and seasonal variations in front position and intensity based on in situ data. We have added related text on lines 55-56 of the revised manuscript and the related in situ data results to the revised Supplementary Materials (Figures S1 to S3).

*Line 103: Can you reference Section 3.4 to provide the magnitude of the wind intensity? When you say relaxation, which values (approx) do you mean.*

The magnitude of the wind intensity is approximately 10 m/s. The relaxation or intensification of the wind indicates a decrease or increase in the wind speed.

*Section 3: How do you demonstrate the dynamics using only SST satellite data? Do you use any other products (satellite, simulations, etc.)?*

The intra-tidal variations of tidal front F1 have been studied by Dong and Guo (2021) using the observation-based tidal current at one point and Equation 5. In this study, we follow the dynamic given by Equation 5 and use the tidal currents from a numerical model that can reproduce the spatial distribution of tidal current in the Seto Inland Sea (Guo et al., 2013) and the satellite SST data to discuss the intra-tidal variations of the other fronts (in Section 4).

*Line 120: Instead of "were ~0.59 and ~-0.16 K," I suggest "were approximately 0.59 K and -0.16 K, respectively."*

Thanks for your suggestion. We have made this change in the revised manuscript.

*Equation 4: Could you provide the ratio of the total number of points for the given period of time to the number of cloud-free valid data points (Nb)? It is important to know this ratio for the representativeness of the samples.*

There are 27576 cloud-free data scenes selected from the entire dataset in seven years, accounting for 45% of the total data. We have added related text on lines 149-150 of the revised manuscript.

*Table 1: Can you add the distance to the stations?*

We have added the distance in Table 1 of the revised manuscript.

*Figure 3: Consider dedicating a full page to this map (and some others, they are small, and the lines are hard to observe. A configuration of 4 rows and 3 columns might improve clarity.*

Thanks. We have changed the configuration of this figure in the revised manuscript.

*Figure 5: Use a consistent format for showing the numbers. Currently, some are rotated to the left, others to the right.*

Thanks. We have revised the figure using a consistent format for the numbers.

*Generally, is much better to use the same colormap for similar representations, many plots have different colormaps.*

Thank you for pointing this out. We have revised these figures in the revised manuscript.

*Line 226: If there is an exception, please specify which ones.*

For the tidal fronts F1, F5, F6, and F8, their intensities were lower during the spring tide than during the neap tide in July. We have added this information on line 233 of the revised manuscript.

*Figure 7: The colormap has changed again. Please ensure consistency.*

Thanks. We have revised this figure.

*Figure 12: Can you explain the difference in values at latitude 34°N?*

The results in the upper and lower areas in this figure are averaged using summer and winter SST data, respectively, so there are differences at the interface in the figure.

*In conclusions I would define the limitations of using only SST.*

Because the satellite data were limited to the sea surface, the vertical structure and its variations of the fronts remain unclear, which requires more field survey data and high-resolution simulations to clarify. We have mentioned this limitation in the last paragraph of the manuscript.

---

## Author Response (AR2)

***Reply to Editor:***

*Thank you for addressing the reviewers' comments. Both reviewers find your revisions adequate and recommend publication. I would like to request a few minor technical corrections before publication (without further review).*

Thanks very much for your helpful comments and suggestions. Our responses to the comments are as follows. The editor's comments are cited in italics.

*Please define the "appearance frequency" in the caption Fig 1b. If this is the same as the front frequency, FF, you can cross-reference Eq. 4.*

Thank you for pointing out this issue. Yes, the "appearance frequency" is the same as the front frequency. We have added a cross-reference to Eq. 4 on line 47 of the revised manuscript.

*Just above Eq.4, you can insert: "The front frequency of each pixel, also referred to as the occurrence frequency of front, was …"*

Thanks for your suggestion. We have made this change in the revised manuscript.

*Typo in Table 1, F8, Akashi (missing i)*

Thanks. We have corrected this typo in the revised manuscript.

*In Fig. 2 and 3 (but also consider for Fig. 4, 7 and 8), the pink lines are not clearly visible. Could you plot a thicker white line behind the pink lines to make them stand out?*

Thanks for your suggestions. For better visibility, we thickened these pink lines and added thinner white lines on top of them in these figures.

*Caption of Figure 15: It would help the reader to describe this figure. You can start with "Evolution of the different terms of Eq.(5) and the tidal current in the cross-front direction for the front F1 during the nine identified phases of the semidiurnal tidal cycle". When you write "...the current phase (...) and the previous phase (...)", current is confused with the ocean current. Please consider rewording as, for example, "at two consecutive phases (solid green line for the selected phase and the dashed green line for the previous phase, for reference)." "*

Thanks. We have revised this caption in the revised manuscript following your suggestions.